# Induction of Multiple Immune Signaling Pathways in *Gryllodes sigillatus* Crickets during Overt Viral Infections

**DOI:** 10.3390/v14122712

**Published:** 2022-12-03

**Authors:** Kristin R. Duffield, Bert Foquet, Judith A. Stasko, John Hunt, Ben M. Sadd, Scott K. Sakaluk, José L. Ramirez

**Affiliations:** 1National Center for Agricultural Utilization Research, Crop BioProtection Research Unit, USDA-ARS, 1815 N. University St., Peoria, IL 61604, USA; 2School of Biological Sciences, Illinois State University, Normal, IL 61761, USA; 3Microscopy Services Laboratory, National Animal Disease Center, USDA-ARS, Ames, IA 50010, USA; 4School of Science, Western Sydney University, Hawkesbury Campus, Richmond, NSW 2753, Australia

**Keywords:** cricket viruses, iridovirus, gene expression, immunity, edible insects, immune tolerance, host-pathogen interactions

## Abstract

Despite decades of focus on crickets (family: Gryllidae) as a popular commodity and model organism, we still know very little about their immune responses to microbial pathogens. Previous studies have measured downstream immune effects (e.g., encapsulation response, circulating hemocytes) following an immune challenge in crickets, but almost none have identified and quantified the expression of immune genes during an active pathogenic infection. Furthermore, the prevalence of covert (i.e., asymptomatic) infections within insect populations is becoming increasingly apparent, yet we do not fully understand the mechanisms that maintain low viral loads. In the present study, we measured the expression of several genes across multiple immune pathways in *Gryllodes sigillatus* crickets with an overt or covert infection of cricket iridovirus (CrIV). Crickets with overt infections had higher relative expression of key pathway component genes across the Toll, Imd, Jak/STAT, and RNAi pathways. These results suggests that crickets can tolerate low viral infections but can mount a robust immune response during an overt CrIV infection. Moreover, this study provides insight into the immune strategy of crickets following viral infection and will aid future studies looking to quantify immune investment and improve resistance to pathogens.

## 1. Introduction

Although the practice of mass-produced insects has been long-standing (e.g., silkworm farming, apiculture, biocontrol agents) [1,2], its application has recently expanded to include uses as livestock and pet feed ingredients [3,4], protein for human consumption [5], chitin for numerous industrial applications [6], and remediation of wastes [7]. Due to the increasing popularity of and demand for insect-based products, there are considerable efforts to maximize insect mass-production [8]. For example, within rearing facilities, microbial pathogens (e.g., viruses, bacteria, fungi) can cause significant mortality, in addition to reducing fecundity and body size. Thus, increasing disease prevention and resistance of insects is essential to the success of this burgeoning industry [9,10,11].

Despite the threat that entomopathogenic infections pose to insect mass-production, we know little about disease prevalence in these settings as systematic screening efforts are currently absent. Crickets (family: Gryllidae) are an especially popular farmed insect; however, they are known to be susceptible to multiple microbial pathogens that can cause disease outbreaks [12,13,14]. For example, the *Acheta domesticus* densovirus (AdDNV), a parvovirus, was implicated to have caused large disease outbreaks in farmed house cricket (*Acheta domesticus*) colonies globally, resulting in wholesale product losses. As a direct response to these outbreaks, many producers switched to farming alternative species, including *Gryllodes sigillatus* in North America, due to reports that they are less susceptible to AdDNV [15]. Furthermore, the number of reports of covert (or, silent) viral infections has increased in reared populations as molecular screening of viruses has expanded to asymptomatic populations [13]. Covert infections by a broad range of microorganisms and other infectious agents are widespread and can manifest as latent infections (e.g., remain within the host cell or integrate into the host genome) or persistent infections with low levels of replication [16]. Importantly, covert infections may become activated resulting in detectable pathology, including mortality.

Beyond agricultural and industrial applications, several species of crickets have long been a model organism within several fields of research (e.g., evolutionary ecology, ecological immunology, insect physiology), where immune effectors have been evaluated across numerous contexts [17,18,19,20,21,22,23,24,25,26,27,28,29]. From these studies, we know that crickets mount an immune response to some microbial pathogens by, for example, exhibiting increased lysozyme-like activity of their hemolymph [30,31], producing a melanization and/or encapsulation response [32,33], and/or increasing circulating hemocytes [34] after controlled exposure to live, inactivated, or simulated (e.g., nylon filaments) pathogens. Although these studies provide insight on the functional downstream outcomes of infection, few have identified and quantified immune gene expression in response to pathogens in these insects [35,36], which is foundational to understanding the molecular basis of their defensive repertoire. Additionally, we do not yet know how crickets respond to naturally acquired pathogenic infections, as almost all assessments have been conducted following controlled inoculation.

Insects possess a suite of cellular and humoral immune defenses in response to viral infection [37] and most of what we know about gene expression as the basis of these defenses comes from work in *Drosophila melanogaster* [38,39], lepidopterans [40], and several mosquito species [41]. Once a pathogen is detected by the insect host, a series of immune signaling pathways are activated intracellularly to respond to infection with a certain degree of specificity, which is in part attributed to the binding specificity of pattern recognition receptors (PRRs) to pathogen associated molecular patterns (PAMPs) (e.g., lipopolysaccharides and peptidoglycans) [42]. The main signaling pathways that mediate immunity in insects are the Toll, Immune deficiency (Imd), and Jak/STAT pathways. Within the Toll pathway, microbes are detected by PRRs that activate the ligand Spätzle, which then binds to Toll receptors and transduces the signals to Cactus-Dif (Dorsal-related immune factor) through a signaling complex containing the adapter MyD88 [43]. Cactus is then cleaved from Dorsal and/or Dif which then translocate into the nucleus and regulate the transcription of effector genes [44]. Within Imd, PRRs recognize invading pathogens and activate the adapter molecule Imd, which activates Relish [45]. Relish is then cleaved and its DNA binding domain translocates into the nucleus where it regulates the transcription of effector molecules [46]. The Jak/STAT pathway is activated as a response to cell stress and/or viral and fungal infection. In this pathway, Domeless is activated and then associated kinases recruit and phosphorylate STAT, which translocates into the nucleus to regulate the expression of downstream effector genes [47]. Additionally, STAT is negatively modulated by PIAS [48]. Activation of these signaling pathways leads to the production of downstream effector molecules, such as antimicrobial peptides (AMPs), that suppress a range of microbes, including viruses [49]. Further, lysozyme is a particularly potent antimicrobial effector in many insects, including crickets [50]; however, the potential antiviral activity of lysozyme has not been widely investigated [51]. Beyond these canonical immune signaling pathways, the RNAi (RNA interference) pathway plays a significant role in the antiviral response in *Drosophila* [52] and has been linked to Jak/STAT, suggesting coordination between these responses [53]. In this pathway, viral dsRNA is recognized by Dicer-2 proteins, which dice it into small siRNAs (small interfering RNAs) which are then loaded into an RNA induced silencing complex (RISC) by Dicer-2 and co-factor R2D2. RISC finds the target transcripts (by complementary sequence with the guide strand of the siRNA) and the Argonaute-2 protein (effector protein of RISC) degrades the target transcript [54].

Recently, we characterized viral abundance in reared *G. sigillatus* crickets from two populations infected with cricket iridovirus (CrIV; family: Iridoviridae): one in which crickets were host to high amounts of viral copies and suffered from increased mortality and reduced fecundity while the other showed no apparent signs of disease and had very few viral copies present [13]. Thus, we consider the diseased population as one exhibiting an overt infection while the healthy population exhibited a covert infection of CrIV. Covert infections of invertebrate iridoviruses are reportedly more prevalent than overt lethal infections in some insect population [55,56,57], which could be due to several factors, including reduced virulence of the virus or increased tolerance of the host. Here, we quantified the expression of immune signaling pathway genes that have been shown to be important in anti-viral immune responses of insects across these two populations of crickets. We selected targets across Toll (MyD88, Cactus, Dorsal, and Dif), Imd (PGRP-LC, Imd, and Relish), and Jak/STAT (Domeless, PIAS, and STAT5B) signaling pathways in addition to a gene encoding for lysozyme. We also measured expression of targets within the RNAi pathway (Dicer-2, R2D2, and Argonaute-2). Finally, because the microbiome has been shown to influence viral dynamics in other insects [58], we quantified the amount of bacteria and fungi present via amplification of the 16S rRNA gene for bacteria and 18S rRNA for fungi to determine if total microbial load plays a role in viral dynamics. We also present TEM images of CrIV viral capsids to confirm active infection within crickets with an overt infection.

## 2. Materials and Methods

### 2.1. Cricket Colonies

*G. sigillatus* crickets were sourced from either of two populations (diseased: a population presenting pathological manifestations of infection, or healthy: an apparently disease-free population) of lab-reared colonies. Symptoms present in the diseased colony were high, intermittent mortality among late-instar nymphs and adults, a strong putrid odor within rearing containers, milky white hemolymph which appeared iridescent under illuminated magnification, increased cuticle and tissue frailty, and underdeveloped or absent ovaries in some adult females [13]. Our previous work found that both populations had the presence of CrIV and AdDNV. Diseased crickets had significantly more CrIV (mean ± 95% C.I. = 3.017 × 10^9^ ± 3.485 × 10^8^ viral copies/µL) compared with healthy crickets (mean ± 95% C.I. = 380.7 ± 131.4 viral copies/µL) [13]. Moreover, diseased crickets also had significantly higher viral loads of AdDNV (mean ± 95% C.I. = 1409 ± 731 viral copies/µL) than healthy crickets (mean ± 95% C.I. = 34.99 ± 18.63 viral copies/µL) [13]. Despite coinfection with CrIV and AdDNV, we concluded that CrIV was the likely disease-causing agent due to the amount of CrIV viral copies present and the apparent symptoms (e.g., iridescent hemolymph). We further determined that the diseased population had an overt CrIV infection while the healthy population had a covert CrIV infection, as the latter had no apparent disease symptoms [13].

These populations were descendants from the same ancestral wild-caught crickets collected from Las Cruces, NM (USA) and have been cultured in a lab setting since 2001. Populations were split and maintained in separate labs since 2007. Rearing methods followed standard cricket rearing protocol within a research laboratory setting [13]. Briefly, about 500 crickets were housed in 55 L plastic storage bins with ventilated lids packed with egg carton to increase rearing surface area. They were provisioned with a standard diet (roughly equal parts Mazuri^®^ Rat & Mouse Diets and Purina^®^ Cat Chow Complete pellets) and water (glass vials plugged with moist cotton) ad libitum. All individuals were housed in an environmental chamber at 32 °C on a 16 h:8 h light:dark cycle. Experimental individuals were at least 1-week-old (no more than 14 days old) post-emergence as adults when they were killed by freezing at −80 °C.

### 2.2. Viral Imaging Via Electron Microscopy

To capture images of virus particles and confirm active infection of CrIV within diseased crickets, we dissected the fat body from a cricket with an overt infection from the diseased colony. Samples were fixed in 2% paraformaldehyde/2.5% glutaraldehyde/0.05 M NaCacodylate/0.005 M CaCl2 (pH = 7). They were post fixed in 1% Osmium for two hours and then processed in graded alcohols, propylene oxide and LX112 (Ladd Research, Williston, VT, USA) resin with 48 h polymerization and shipped to the USDA-ARS Microscopy Services Laboratory at the National Animal Disease Center (Ames, Iowa, United States) for further processing. Thick sections (1 µm) were performed on select samples and a toluidine blue stain and basic fucsin stain were applied. Polaroid photos were taken of these images and the area of interest for thin sections was identified. A uranyl acetate and Reynold’s lead stain were performed on the thin section before being examined with a ThermoFisher FEI Tecnai G^2^ BioTWIN electron microscope (ThermoFisher FEI Co., Hillsboro, OR, USA) and images were taken with a Nanosprint12 camera (AMT Corp., Woburn, MA, USA) [59].

### 2.3. RNA Extraction and cDNA Synthesis

We extracted RNA from whole body homogenates of individual crickets from each population (20 crickets/population). Previously frozen (−80 °C) crickets were placed individually in tubes with 1 mL sterile 1x PBS (pH 7.2) and two 3.2 mm diameter sterile stainless-steel beads and macerated using a TissueLyser II (Qiagen, Hilden, Germany). The resulting liquid homogenate was removed (about 0.9 mL) and placed in a new sterile tube for RNA extraction. RNA was extracted from 100 µL of cricket homogenate using the RNeasy Mini prep kit (Qiagen) following the “Purification of Total RNA from Animal Tissues” protocol. The concentration of RNA within each sample was estimated via a NanoDrop One^C^ Microvolume UV-Vis Spectrophotometer (ThermoFisher, Waltham, MA, USA) and 260/280 and 260/230 values were above 1.8 for all samples. RNA from each sample was diluted to 1 µg, treated with DNA Wipeout, and then converted to cDNA using the QuantiTect Reverse Transcription Kit (Qiagen) prior to conducting reverse transcriptase quantitative PCR assays. All cDNA samples were stored at −20 °C until further use.

### 2.4. Gene Target-Specific Primer Design

We searched for the target genes in the head transcriptome of *G. sigillatus* [60] using the BLAST+ command-line application [61,62,63]. Specifically, protein sequences from the mosquito *Aedes aegypti* or nucleotide sequences of the cricket *A. domesticus* were used as a reference, using blastx and megablast, respectively with default settings to find *G. sigillatus* sequences. The resulting cricket sequences were blasted to the nr database to confirm gene identity [64,65], and sequences were manually trimmed to remove potential chimeric sequences. Subsequently, the coding regions of these sequences were translated to their respective proteins in Ugene [66], and were aligned with homologous proteins from a representative set of insects (the moth *Bombyx mori*, the bee *Apis mellifera*, the fruit fly *Drosophila melanogaster*, the mosquito *Aedes aegypti*, the beetle *Tribolium castaneum*, the termite *Zootermopsis nevadensis*, the grasshoppers *Locusta migratoria* and *Schistocerca gregaria*, and the cricket *Gryllus bimaculatus*), as available on the ncbi protein database [65]. Alignments were performed and visualized in Ugene using the MUSCLE algorithm [67] with default settings (Appendix A). For Dicer-2, Argonaute-2, Relish, Dorsal and Dif, phylogenetic trees were made to further confirm sequence identity (Appendix A). Protein sequences were aligned using MUSCLE with default settings on a linux machine, and Maximum Likelihood trees were subsequently made using RAxML v. 8.2.12 [68]. Trees were visualized using the Interactive Tree of Life (iTOL) [69]. All sequences were deposited in GenBank (see Table 1 for accession numbers).

All primers used in the present study were designed using Primer-BLAST (NCBI) (all primers from IDT, Inc., Coralville, IA, USA). For quantification purposes, we designed primers targeting the 18S and 16S ribosomal RNA (Table 1) as invariant housekeeping genes. These were selected based on their performance/ranking via RefFinder [70,71], which uses the algorithm from major computational programs such as geNorm, Normfinder, and Best-Keeper to compare and rank candidate reference genes. We then calculated the geometric mean for expression of these two genes for each individual and used this as our reference target for gene expression. To evaluate whether fungal or bacterial load could contribute to differential viral loads between populations, we quantified the 16s rRNA for bacteria (16SrRNA-Fw 5′-TCCTACGGGAGGCAGCAGT-3′ and 16SrRNA-Rv 5′ GGACTACCAGGGTATCTAATCCTGTT-3′) and the 18s rRNA (18SrRNA-Fw 5′-AGATACCGTCGTAGTCTTAACCATAAACT-3′ and 18SrRNA-Rv 5′-TTCAGCCTTGCGACCATACT-3′) for fungi from crickets across both populations.

### 2.5. Reverse Transcriptase Quantitative PCR (RT-qPCR) Detection and Quantification

RT-qPCR reactions were run on a Quant-Studio 6 Real-Time PCR instrument (Thermo Fisher Scientific, Waltham, MA, USA), and included a melt-curve stage to confirm product specificity. One microliter of cDNA product was used in a 10 µL RT-qPCR reaction using gene specific primers (Table 1) and PowerUp SYBR green Master mix kit (Qiagen). RT-qPCR cycling conditions consisted of holding at 50 °C for 2 min and 95 °C for 2 min and 40 cycles of 1 s at 95 °C and 30 min at 60 °C. The identities of targets were confirmed by mapping sequences to the reference target genes using default settings in Geneious Prime^®^ following Sanger sequencing.

### 2.6. Statistical Analysis

Gene expression profiles were evaluated using the ΔΔCt method [72]. We used an unpaired *t*-test with Welch’s correction for each immune gene target to compare expression of healthy and diseased crickets between populations, in addition to comparing microbial loads. All expression data were log_2_-transformed to fit normality assumptions. All analyses and graphs were performed and created using GraphPad Prism 9 (version 9.0.0).

## 3. Results

### 3.1. TEM Imaging

TEM analysis revealed substantial quantities of virions and confirmed the presence of large (~160 nm) icosahedral viruses in cells from the dissected fat body of a cricket from the diseased population (Figure 1). Cricket cells were packed with virions that often formed small paracrystalline arrays (e.g., Figure 1C).

### 3.2. Immune Signaling Pathways

Within the Toll pathway, the relative expression for the transcription factors *Dorsal* (Welch’s corrected *t*(26.06) = 3.626, *p* = 0.0012) and *Dif* (Welch’s corrected *t*(27.50) = 5.779, *p* < 0.0001) were significantly higher in the diseased population than in the healthy population. Similarly, the expression of *Cactus*, a negative regulator, was higher in diseased crickets (Welch’s corrected *t*(22.26) = 4.058, *p* = 0.0005). The adapter molecule *MyD88* did not differ between populations (Welch’s corrected *t*(35.32) = 1.285, *p* = 0.207; Figure 2A).

For targets within the Imd pathway, both the transcription factor *Relish* (Welch’s corrected *t*(19.95) = 6.361, *p* < 0.0001) and the adapter molecule *Imd* (Welch’s corrected *t*(27.25) = 3.507, *p* = 0.0016) were more highly expressed in the diseased population. However, expression of the pathogen recognition receptor *PGRP-LC* was similar across populations (Welch’s corrected *t*(28.84) = 0.6128, *p* = 0.5448).

Finally, the relative expression of all targets measured in the Jak/STAT pathway were significantly higher in the diseased population than in the healthy population, including the receptor *Domeless* (Welch’s corrected *t*(35.31) = 5.525, *p* < 0.0001), the transcription factor *STAT5B* (Welch’s corrected *t*(24.21) = 4.728, *p* < 0.0001), and the negative regulator *PIAS* (Welch’s corrected *t*(24.49) = 5.424, *p* < 0.0001) (Figure 2C).

### 3.3. RNAi Pathway

All targets measured within the RNAi pathway, *Argonaute-2* (Welch’s corrected *t*(19.62) = 9.101, *p* < 0.0001), *R2D2* (Welch’s corrected *t*(19.39) = 5.933, *p* < 0.0001), and *Dicer-2* (Welch’s corrected *t*(27.77) = 11.19, *p* < 0.0001), were more highly expressed in the diseased population than in the healthy population (Figure 3A).

### 3.4. Lysozyme

The relative expression of lysozyme (Welch’s corrected *t*(16.03) = 7.504, *p* < 0.0001; Figure 3B) was significantly higher in the diseased population compared with the healthy population.

### 3.5. Microbial Load

Both fungal load (Welch’s corrected *t*(28.85) = 1.152, *p* = 0.2588) and bacterial load (Welch’s corrected *t*(68.00) = 1.348, *p* = 0.1821) were similar between healthy and diseased populations based on relative quantification of fungal 18s rRNA and bacterial 16s rRNA, respectively (Figure 4).

## 4. Discussion

Despite its importance in host response to pathogens, we still know little about the molecular basis of cricket immunity, and even less about immune responses within the context of covert infections. To improve our understanding of host–virus interactions in crickets, we evaluated canonical immune signaling pathways that have been shown in other arthropod systems to be involved in immunity to microbial organisms, including viruses. By quantifying gene expression across several facets of the invertebrate immune system, we have begun to characterize the immune response to overt cricket iridovirus (CrIV) infections in the popularly reared *G. sigillatus* cricket. Specifically, we found that crickets infected with high levels of cricket iridovirus had higher gene expression across the Toll, Imd, and Jak/STAT immune signaling pathways as well as within the RNAi pathway.

Activation of the Toll and Imd pathways are typically associated with defenses against pathogenic fungi, bacteria, and protozoa. Furthermore, these two pathways have been linked with the antiviral response in Diptera [73,74], but less is known about their role in orthopterans. Both Jak/STAT and RNAi pathways are known to play important roles in antiviral immunity and so it is not surprising that targets across these two pathways were upregulated in crickets with large CrIV viral loads in our study. Our study also evaluated an important antimicrobial effector, lysozyme, which has been found to have antiviral activity against dengue virus in mosquitoes [75] as well as against other viruses infecting eukaryotic hosts [51]. Our transcript level analyses show that the population of diseased crickets had significantly higher expression of lysozyme compared with the healthy population. This suggests that lysozyme might play a significant role in the cricket’s efforts to control the systemic replication of CrIV. Taken together, we can conclude that crickets with overt CrIV infection have an immune profile exhibiting strong induction of critical immune pathway components across Toll, Imd, Jak/STAT, and RNAi. At the same time, it begs the question of whether crickets tolerate viruses when they occur at a lower concentration and fully engage the immune system only when viral loads surpass a certain threshold. Unfortunately, our study is unable to fully answer this question given that our control (healthy population) also carried CrIV, albeit at significantly lower levels.

A few previous studies have identified immune related genes in Orthoptera, including crickets; however, none to our knowledge have investigated an antiviral response in this order. An enzyme similar to the AMP prolixicin was discovered in *A. domesticus* suspected of being infected with a gregarine parasite and found to occur at higher concentrations in juveniles compared with adults [76]. In the black field cricket (*Teleogryllus emma*), researchers identified 58 differentially expressed unigenes and several AMPs following inoculation with *E. coli* [77]. In *Gryllus bimaculatus*, 4 inducible lysozymes and 6 AMPs were identified with similarities to defensin and diptericin, as well as pyrrhocoricin, prolixicin, and hemiptericin [78]. A comparative transcriptomic analysis of the immune response of migratory locusts challenged with *Metarhizium* fungi identified immune related unigenes including those involved with Toll, Imd, and Jak/STAT pathways, with 58 and 3 differentially expressed in the fat body and hemocytes, respectively [79]. It also found higher expression of lysozyme transcripts post-infection. While it is difficult to draw direct comparisons across these few studies, some patterns emerge, including the roles that the canonical immune signaling pathways and their effectors play in the Orthopteran immune system.

Our findings add to the current understanding of the insect host immune response to iridovirus in crickets. Previous work has shown that lab reared *Gryllus texensis* crickets infected with an iridovirus have significantly lower phenoloxidase activity than uninfected crickets [80]. This contrasts with our findings of increased immune gene expression, but we did not assess any genes involved directly in the phenoloxidase cascade and phenoloxidase activity has been shown to be negatively associated with other components of immunity or reduced following immune activation in other insects [81]. Although, to our knowledge, there are no reports of studies that have investigated molecular markers of immune activation following an infection with CrIV, several studies have characterized host response to the closely related Invertebrate Iridescent virus 6 (IIV-6) in *Drosophila* [13,82]. From these, we know that the IIV-6 genome encodes for proteins that can inhibit insect host immune responses, including RNA silencing by the RNAi pathway (e.g., 340 L) [83], which is the primary defense against IIV-6 [84,85]. In our study, both Dicer-2 and Argonaute-2 were upregulated in crickets with overt CrIV infections, suggesting that the RNAi pathway is also important in antiviral defense for *G. sigillatus*. IIV-6 was also found to be able to inhibit both Imd and Toll pathways [86]. Interestingly, while there is no evidence that the Jak/STAT pathway confers immunity against IIV-6 infection in *Drosophila* [85], our study indicates significant induction of Jak/STAT pathway components in response to CrIV. Importantly, we found no evidence of viral inhibition of these responses at the transcriptional level in the present study.

Why individuals from one population suffer from overt CrIV infections while the other maintains covert, asymptomatic infections remains an open question. One possibility is that the diseased population was exposed to an undetected microbe (e.g., bacteria or fungi) that made it more susceptible to an overt viral infection. Indeed, previous studies have demonstrated that co-infection with IIV-6 and a Gram-negative bacterium result in more rapid mortality in *Drosophila* [86]. Although we did not find differences in total microbial loads (Figure 4), we did not characterize microbiomes and therefore cannot rule out the possibility that bacterial or fungal composition are different between populations. Future studies will characterize the microbiome between populations with overt and covert infections to determine if the microbiome may play a role in promoting (or inhibiting) overt infections. While we previously found evidence of low viral loads of AdDNV in both populations of crickets [13], AdDNV has not previously been associated with disease in *G. sigillatus* [15]. Still, we did find that crickets with overt infections of CrIV had significantly higher (albeit relatively low) AdDNV viral loads. Additional studies will evaluate the impact of viral dynamics on infection outcomes. Another possibility is that intrinsic (e.g., inbreeding) or extrinsic (e.g., rearing environment) factors may impair immune function leading to the opportunistic reactivation of covert infections [87]. However, crickets with high levels of CrIV were able to mount an immune response across multiple pathways and so it is unlikely that crickets from the diseased population have a dysfunctional immune response, at least at the transcription level. Further studies probing post-transcriptional and post-translational outcomes will be essential to understanding the role immunity plays in regulating viral loads.

In the present study, we assessed genes that are known to be important in immunity in other model insects (e.g., mosquitos, flies, moths). Future studies (e.g., RNAi knockdown experiments) will determine the importance of specific pathways on clearing or decreasing viral loads in these crickets. Furthermore, a comparison of complete transcriptomes between populations is required to obtain the global gene expression repertoire of infected crickets. These studies will be essential to fully characterize the defensive strategy of crickets at the transcript level and will improve our understanding of how crickets can tolerate low levels of CrIV and maintain covert infections. Hampering these efforts is the fact that few genetic resources for crickets currently exist [88], including the absence of a complete and annotated genome of *G. sigillatus*. Advancements in this field will greatly aid research efforts, including those seeking to improve production of reared beneficial insects.

## 5. Conclusions

By evaluating the induction of immune-related genes across populations of crickets with an overt or covert infection, we can begin to understand the immune responses of *G. sigillatus*, a popularly reared cricket and a model insect across multiple branches of research. Crickets with an overt infection of the highly pathogenic CrIV presented significantly higher induction of multiple genes across all canonical immune signaling pathways, in addition to the RNAi pathway, compared to crickets with a covert infection. Our data suggest that *G. sigillatus* can tolerate low levels of viral infection and are able to mount an immune response when faced with an overt viral infection.

## Figures and Tables

**Figure 1 viruses-14-02712-f001:**
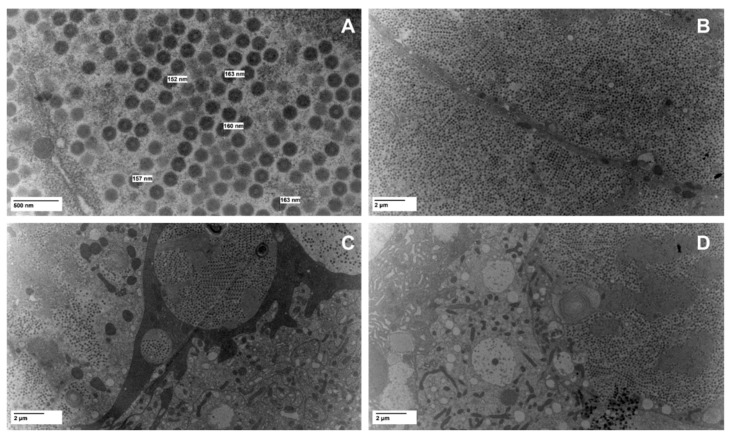
TEM images of fat body tissues dissected from a diseased cricket infected with cricket iridovirus (CrIV) at (**A**) 30,000× magnification and (**B**–**D**) 4800× magnification.

**Figure 2 viruses-14-02712-f002:**
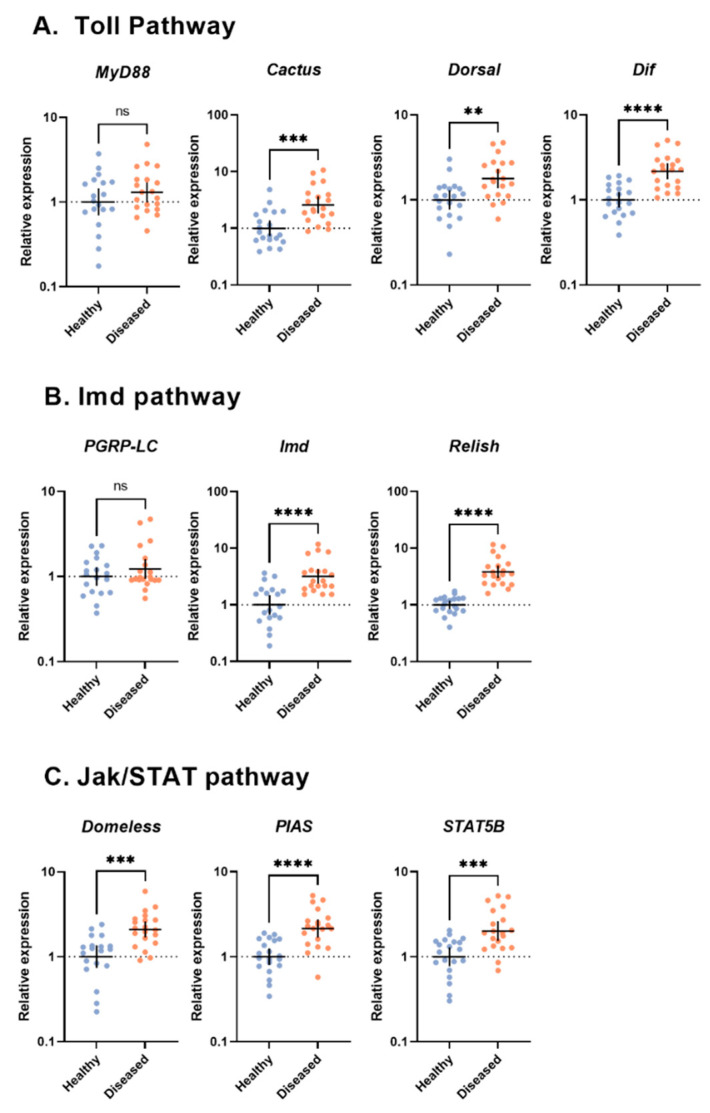
Gene expression profiles of immune signaling pathways (**A**) Toll, (**B**) Imd, and (**C**) Jak/STAT in the whole bodies of *Gryllodes sigillatus* adults from two populations (sample size for *MyD88*, *Domeless*, and *PGRP-LC*: diseased n = 20 and healthy n = 19; sample size for *Cactus*, *Dorsal*, *Dif*, *Imd*, *Relish*, *PIAS*, and *STAT5B*: diseased n = 20, healthy n = 20). Each dot represents a single cricket with horizontal lines representing mean expression with 95% confidence intervals. The statistical significance of fold change values was determined on log^2^-transformed values via unpaired *t*-test with Welch’s correction. ** *p* < 0.01, *** *p* < 0.001, **** *p* < 0.0001, ns = not significant.

**Figure 3 viruses-14-02712-f003:**
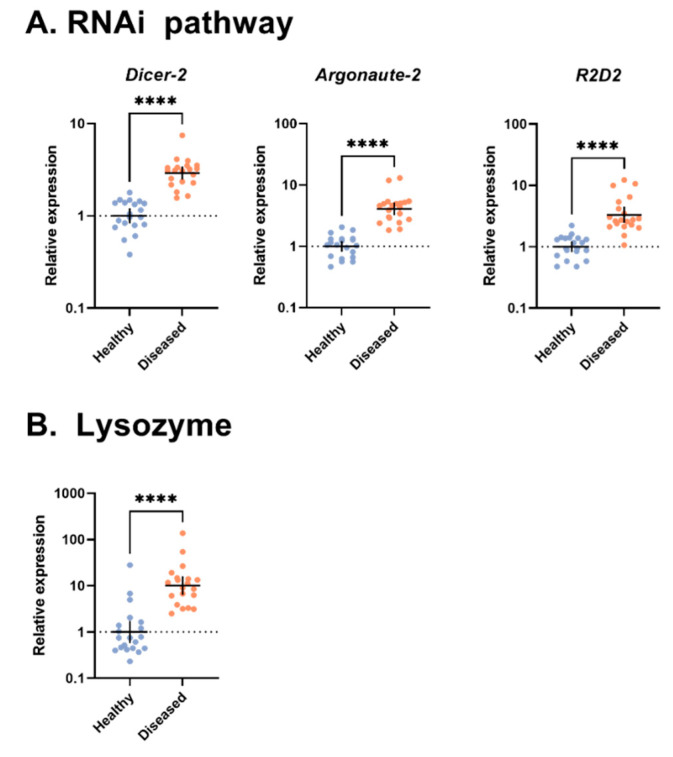
Gene expression profiles of (**A**) the RNAi pathway and (**B**) lysozyme in the whole bodies of *Gryllodes sigillatus* adults from two populations (sample size: diseased n = 20, healthy n = 20). Each dot represents a single cricket with horizontal lines representing mean expression with 95% confidence intervals. The statistical significance of fold change values was determined on log^2^-transformed values via unpaired *t*-test with Welch’s correction. **** *p* < 0.0001.

**Figure 4 viruses-14-02712-f004:**
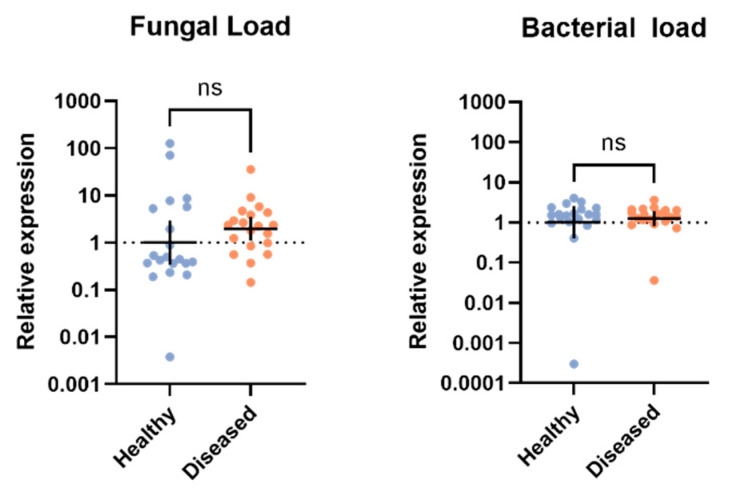
Fungal and bacterial loads via relative quantification of fungal 18s rRNA and bacterial 16s rRNA, respectively, in the whole bodies of *Gryllodes sigillatus* adults from two populations (sample size: diseased n = 20, healthy n = 20). Each dot represents the microbiocidal load value from a single cricket and the horizontal black bar indicates the mean microbial load with 95% confidence intervals. The statistical significance of fold change values was determined on log^2^-transformed values via unpaired *t*-test with Welch’s correction; ns = not significant.

**Table 1 viruses-14-02712-t001:** Primers used to quantify gene expression in *Gryllodes sigillatus* adults in this study.

Target	Gene ID	Primer Sequence	Amplicon Size (bp)	Efficiency	R^2^
*Dorsal*	ON081012	GGTAGGGGCTCTCTTTGGTC	107	98%	0.9978
		CGTTCTGCTGGCTCTATTCC			
*Dif*	ON081011	TATGAATGCGAAGGGAGGTC	130	98%	0.9963
		ACAGCACGACCCTGATAACC			
*Cactus*	ON081013	GTGTGACCAGCGTAAGTGGA	75	92%	0.9979
		CCTCAGCAGTGTGTTGCATT			
*MyD88*	ON081014	AACGGCTCCAGCATCTAAAA	115	90%	0.9949
		TGGTGGATCTGTCAAGCAAG			
*PGRP-LC*	ON081023	AATAGCCAGAGGAGCAGCAA	99	100%	0.9982
		GGCCAAACTGGAGATACCAA			
*Imd*	ON081024	ATTCCTCGCATCAACACTCC	143	96%	0.9839
		TCAGGTGATGGTGATTTGGA			
*Relish*	ON081022	GGCAGTTTCACCTTCCACAT	118	96%	0.9999
		GCTGCAGATGGCTCTAAAGG			
*STAT5B*	ON081015	GCCCCATACCATGTCCTAGA	109	91%	0.9971
		TATGTGCACAATCCCCTCAA			
*PIAS*	ON081016	GGTCACAAAGCCTTCAGGAG	82	100%	0.9973
		AGTTCTCTGGACGTGCCAAT			
*Domeless*	ON081017	CCATTCAGGCACCAGAAGAT	124	99%	0.9995
		TGCCAAAAGAACCAGTTTCC			
*Argonaute-2*	ON081018	TGCATGTTCATCCCTTGAAA	135	95%	0.9976
		GTTCCCGGCAAGACATTAAA			
*Dicer-2*	ON081020	CCCTTTCTCCATGACTTCCA	78	100%	0.9992
		CCTCCAATTTTCAGCACCAC			
*R2D2*	ON081019	ATGTCTGCCTGTTGGGAAAC	99	99%	0.9986
		GCGCTCACGTGTACTGTTGT			
*Lysozyme*	ON081021	TTACGACTACGGCCTGTTCC	84	98%	0.9994
		TCGCACTTCATCTTGCAATC			
*18S rRNA*	KR904053	GCCGTTCTTAGTTCGTGGAG	130	97%	0.9979
		CGCCTGTCCCTCTAAGAAGA			
*16S rRNA*	AF514593	TCGTCACCCCAACCAAATAC	106	96%	0.9984
		TAATGGGGGACGAGAAGACC			

## Data Availability

The datasets generated during the current study will be available in the GenBank repository upon acceptance.

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
