# Peer review of "Induction of Multiple Immune Signaling Pathways in Gryllodes sigillatus Crickets during Overt Viral Infections"

_viruses, 2022, doi:10.3390/v14122712_

Round 1

Reviewer 1 Report

Comments to Authors

            This paper represents one of the few that explores the immune response of a cricket to a virus.  Little work on the molecular level has been done on the immunology of this large group of insects.  Crickets are an important model system in behavioural ecology, as well as a growing source of farmed protein. 

            As the authors note, little previous work exists on this group of insects.  Given that, it is important that the qPCR follow the MIQE guidelines (see https://www.bio-rad.com/en-ca/applications-technologies/miqe-rdml-guidelines?ID=LUSO8N4EH).  The most important of these we’ve listed below. 

1.     Did you remove the gut from the crickets before homogenizing them to extract RNA?  These are the likely source of the bacterial RNA.  This issue should be discussed in the paper – i.e. how the retention of the gut complicates and conclusions based on bacterial count.

2.     Ribosomal genes are poor choices for normalization because they occur in far higher concentrations than other genes.  Provide justification for this choice. 

3.     Did you use normfinder, or another method, to demonstrate the stability of your reference genes?  If so, add this information. 

4.     There was no ‘healthy’ population as both were infected.  Perhaps fair to say differences between relative levels of infection. 

5.     Line 292.  Or lysozyme in response to bacteria.  

6.     Paragraph, Line 338.  Also need to discuss that the location of the bacteria will influence its impact on the immune system, (i.e. gut or  hemocoel).  Unfortunately this cannot be differentiated when using whole homogenates, but the issue should be discussed.

7.     Please ensure that you follow MIQE guidelines in reporting your qPCR data.  For example,

a)     There is no mention of testing for purity. If you used a nanodrop, confirm that each sample had a purity rated 1.8 or greater for each sample.   

b)    There is no mention of integrity checks.  These are critical.  Did you use a Bioanalyzer or other method?  Describe these methods and the results. 

c)     The primer table should include the efficiencies. 

d)    In a supplemental file, provide confirmation of that the appropriate standard curves and thermal gradients were used. 

e)    Provide confirmation that at least one representative of each amplification sequence per target was sent away for sequencing.  Melt curves only represent specificity if you’ve confirmed the sequence yourself.

Reviewer 2 Report

In this paper, 14 immune-related genes, belonging to 6 insect immune pathways, are identified in Gryllodes sigillatus crickets. Their expression is then measured by qPCR and compared between two populations of crickets, which present different types of infections by DNV (covert vs active). The main finding is that the expression of these immune genes is higher in the population with an active viral infection than in the one with a covert infection. This is a descriptive manuscript, that attempts to contribute to the understanding of antiviral immune pathways of orthopteran insects. Major concerns should be addressed before publication:

- The authors mention that two viruses are present in the populations, but don’t show any data on this. What evidence do the authors have to claim that IV and DNV are present in both these cricket populations?

- It is stated that one population has an active infection by the IV, while the other population has a covert infection:

          _ What is the difference between a covert and an active infection? Is the virus replicating during a covert infection? If so, isn’t it also an active infection? Regarding insect literature, the terms persistent vs acute infections are often used. Can these be applied here, to avoid confusion? If this is not possible (due to different meanings), I suggest that the authors explain in the introduction the difference between the two types of infections (symptoms vs non-symptoms; virus replicating vs non-replicating; high viral levels vs low viral levels; …)

          _ What evidence do the authors have to support that one population has an active IV infection while the other has a covert IV infection? In the manuscript, this is supported by Figure 1, where IV virions are identified by TEM in the crickets with active infection; however, there are no pictures from the population with a covert infection; did the authors observe that in the crickets with a covert infection, no virions could be identified? And is the virus still identifiable in the population with a covert infection? Can this be complemented with PCR (or qPCR) data, demonstrating different levels of this virus in both populations? Comparison of DNV levels is also important, so that the levels of these two viruses can be co-related with the remaining results.

- The transcripts of interest were found via a BLAST search, using as query sequences from other insects. However, the identity of these sequences should be further confirmed by phylogenetic analysis and protein domain prediction.

- A dicer sequence was identified and then quantified by qPCR. However, insects are known to have two dicer proteins: dicer1, effector of the miRNA pathway; and dicer2, effector of the siRNA pathway, the canonical antiviral RNAi pathway. Which of these was used? I believe that it is critical to present a phylogenetic analysis to confirm the identity of this dicer sequence.

- Regarding the argonaute sequence, the situation is even slightly more complex. The Argonaute superfamily has 2 clades of proteins, the AGO and the PIWI. Insects have two argonautes belonging to the AGO clade, namely argonaute1 (miRNA pathway) and argonaute2 (siRNA pathway). In addition, insects have a species-specific number of PIWI proteins. Once again, a protein domain prediction and a phylogenetic analysis is critical to confirm the identity of this argonaute sequence.

- Were the qPCR primers tested for efficiency?

- If I understand correctly, two reference genes were used, namely the 18S and the 16S ribosomal protein genes. How were these genes selected? Was their stability tested in advance?

- Still regarding the reference genes, it is advisable that the used reference genes are not involved in the same cellular pathways, because this can result in biased results. Therefore, an extra reference gene should be used, that is not a ribosomal protein. The reference genes should be selected based on stability of expression between the two groups.

Small comments:

- In the introduction, the authors mention that the majority of knowledge on insect immunity comes from research in D. melanogaster; while true, quite some research has been done in other species, namely in lepidopterans; so I suggest that some mains of other insects are  added here.

- Line 72-95: the authors start with “The main signalling pathways that mediate antiviral immunity in insects are the Tool, Imd and Jak/STAT pathways”, and then continue by describing the different pathways. However, not only antiviral mechanisms are described in continuation, but also anti-bacterial and anti-fungal pathways, which induces some confusion and makes it difficult to read. Therefore, I suggest that this section of text is restructured.

- Lines 91-95: the description of the RNAi antiviral response is not accurate; “In this pathway, viral dsRNA is recognized by Dicer proteins and are loaded (WHAT IS LOADED?) into a pre-RNA induced silencing complex (RNA-INDUCED SILENCING COMPLEX – RISC) where they (THEY, WHO?) are degraded by the RNase activity of Argonaute.”; viral dsRNA is recognized by dicer2, which dices it into siRNAs (small interfering RNAs); these siRNAs are loaded into an RNA-induced silencing complex (RISC); this complex finds the target transcripts (by complementary with the guide strand of the siRNA) and the effector protein of RISC, the Argonaute2 protein, degrades the target transcript.

- Line 326: “Although there are currently no known studies that have investigated molecular markers…” should be replaced by something in the line of “Although, to our knowledge, there are no reports of…”

- Line 330: you mention that IIV-6 can inhibit RNAi; does this virus encode a viral suppressor of RNAi (VSR)?

- Line 356-357: “we targeted’ should be replaced by “we studied/assessed,…”; “(e.g., mosquitos)”, but also flies, moths, …

- Indicate the number of samples per each measurement (if pools, indicate the number of animals per pool)

-Line 198: Real-time quantitative PCR (RT-qPCR)

Round 2

Reviewer 2 Report

Dear authors,

Thank you for your replies. The new version of the manuscript already shows improvement. However, some concerns remain to be addressed:

- Regarding the reference genes, I would like to again point that genes belonging to the same pathway/class should not be used. The fact that the two ribosomal proteins come out from the RefFinder analysis as more stable is likely due to the fact that they are co-regulated. And this will result in biased quantifications. See, for instance, DOI 10.1007/978-1-4939-0733-5_3

“It is important to avoid selecting multiple genes from the same pathway or functional class because strong coregulation may interfere with proper analysis of expression stability.”

Thus, the authors should re-do this analysis (possibly including the tubulin gene that was already measured).

I insist in this since the entire manuscript is based on qPCR results. In addition, all the measured factors show the same trend (higher in the diseased population). So it is important to exclude that this is an artifact introduced by the reference genes levels.

- Thank you for pointing out the explanation regarding the IV and DNV infections (lines 107-109 of Introduction; lines 150-160 of M&M):

o    I think the manuscript would benefit from emphasizing this in the results section, maybe before the TEM results?

o   In the discussion, it’s important to mention that virions have not been detected in the healthy population, which is in line with the fact that lower viral levels are detected in these animals (and thus likely lower levels of virions); but that we can be confident that these animals are infected since the viral transcripts are detectable via qPCR.

-       - In the discussion, the fact that the diseased population is also infected by DNV should not be overlooked – please add/discuss

-    - You discuss that other pathogens can differ between populations, as for instance bacteria. In this context, it is also relevant to discuss that other factors might differ - a particularly interesting example to mention is a possible fungal infection since lysozyme has been shown to be upregulated in locusts infected with Metarhizium; in addition, it should be also mentioned that the populations have been kept in different labs – although similar breeding conditions have been used, it’s not possible to completely exclude unexpected differences

Small:

-        Lines 51-54: rephrase: “Covert infections by a broad range of microorganisms and other infectious agents are known to be widespread and… “  

-        Lines 267-263 have been deleted by mistake?

-        Figures 2 and 3 appear twice

-        Caption of Figure 3: A) RNAi; B Pvr; C) lysozyme
